# Characterising Communication of Scientific Concepts in Student-Generated Digital Products

## Helen Georgiou 

School of Education, University of Wollongong, Wollongong 2522, Australia; helengeo@uow.edu.au

**Abstract:** New assessment types that include multimodal and digital elements are increasingly being used to assess university students' 'soft skills' such as communication, as well as their science content knowledge. However, very little is known about how or how well such products assess communicative elements, particularly when these elements are so intricately linked with disciplinary knowledge. This paper presents a novel way of looking at these new digital assessments in science. Using semantic density, a concept from the framework of Legitimation Code Theory (LCT) that conceptualises complexity, we consider how to characterise learners' communication of complex science in the digital products. Results show that successful products 'negotiate' complexity in distinct ways and that language and image work together in the products to build meaning. This approach is a 'first step' in characterising discipline-based communication skills through the development of a preliminary conceptual framework that will inform pedagogies and assessment practices surrounding student-generated digital products, in an effort to improve outcomes for science students.

**Keywords:** student-generated digital products; science communication; Legitimation Code Theory; multiple representations; assessment

## 1. Introduction

Relatively new Information and Communications Technologies (ICTs), such as podcasts, videos and simulations are increasingly appearing as assessments in university science subjects [1,2]. There seem to be two drivers for this. First, developments in ICTs mean learners can create new, more complex, (usually dynamic) media forms more efficiently and with relatively user-friendly, low-cost software [3] and second is the push for university graduates to have a set of 'soft skills' in addition to 'content knowledge' in order to be ready for 21st-century jobs [4]. In terms of communication, for instance, recent reforms related to tertiary science education in Australia explicitly stipulate that graduates must be able to communicate 'to a range of audiences', 'for a range of purposes' and 'using a variety of modes' [5] by the end of their degrees in order to be prepared for diverse employment outcomes [6–8].

These assessments are new territory because although traditional university assessments for science subjects usually incorporate a range of representations, these are often in a static form, like a poster, or mediated by the learner, such as in a multimodal presentation [9]. Rather, the digital products discussed in this paper are both dynamic and standalone (can be played from beginning to end). A review of the use of the 'Dynamic Standalone Product' (DSP) in university science subjects reveals that they take on a variety of forms, with the most common including three or four main modes: image, narration, text on screen (labelling) and either animation or video [10]. Such a resource might resemble a PowerPoint presentation with overlaid narration that plays from beginning to end, or a short film which includes different scenes and role play. For a discussion on naming conventions and an outline of different types of digital products, see [11]. There is sufficient evidence to suggest constructing a DSP is advantageous for learners in terms of developing disciplinary knowledge,

increasing engagement, facilitating collaborative skills (when working in groups) and developing communication skills, e.g., [12–14].

For example, Hoban et al. [12] found that constructing a 'slowmation' (a type of DSP that involves compiling images together to create a 'slow' animation) helped to rectify alternative conceptions related to the phases of the moon. In [14], Mills et al. find that engagement in the slowmation task improves interest in science. Further, it is argued that the dynamic and multi-representational elements of these resources might be uniquely responsible for these potential benefits [14].

To understand why DSPs have such potential, we might look to the research on representations in science education. A strong link between disciplinary understanding, epistemology and communication skills at all levels of education is identifiable in this literature, e.g., [15–23]. That is, a range of studies show that in working with language and other representations, students develop communication skills, improve conceptual understanding and are likely to gain a more sophisticated understanding of the nature and structure of scientific knowledge. For example, Pelger and Nilsson found, in their study, that intentional instruction on *communication* techniques actually improved subject understanding amongst university science students [21]. These benefits have been found at all educational stages. For instance, a study on primary aged students shows that the use of a multimodal task as an assessment improved expression, encouraged refinement of thinking and developed knowledge about the topic [20]. When ninth graders were asked to explain the 'work-energy' concept using a range of representations, students gained both quantitative understanding and greater epistemological awareness [22]. Klein and Kirkpatrick explain that this might be because '( . . . ) representations do not simply transmit scientific information; they are integral to reasoning about scientific phenomena' [23].

It seems natural to use technology to facilitate working with representations. However, there is also no guarantee that the potential benefits will be realised. Research on the use of technology more generally (in science education) has established that beneficial or successful integration depends on a variety of complex factors [24,25]. However, most of this kind of research primarily focuses on the use of technology by the instructor, so what we know about student-created products is even more limited. For instance, in the multimedia literature, Cognitive Load Theory is used to justify elements of multimedia design for instructors, such as avoiding irrelevant information and combining visual and auditory modalities [26]. However, this field of research does not provide a basis for how knowledge is represented, by a maker, in a multimodal assessment product. Thus, we know little about the neophyte scientist and communicator; what do they understand about multimodal science communication? How is this reflected in the product? How does this understanding develop? Reyna and Meier [11], in their review of the field, state that the field of Learner Generated Digital Media (LGDM) is still very much in its infancy. They identify a lack of theoretical underpinning, which leads to variable results when DSPs are used as assessments. The lack of understanding of the principles that underlie effective digital products is problematic because without them, it is often left to the subjective views of individual instructors judging the overall 'feel' of the product and this limits what can be gained from the process (p. 102).

In order to understand the relationship between representations, communication and knowledge, we developed a preliminary theoretical framework from the close analysis of two DSPs, which will ultimately help develop our understanding of principles of dynamic media creation in order to inform related pedagogies and assessment practices. Because our particular interest is the intersection between knowledge and communication, draw on a sociologically-based theory, known as 'Legitimation Code Theory', which is designed to decode knowledge practices.

Legitimation Code Theory (LCT) is a sociologically-based theoretical framework grounded in the social realist philosophy and focused on knowledge [27,28]. This means that it considers knowledge as a central consideration when considering social practices (such as education). Maton emphasises that this is a point of difference in education research, which is 'knowledge blind' and where psychologically-based approaches, such as Cognitive Load Theory, focus on what is happening 'in the mind', whilst other sociologically-based approaches focus instead on power relations, both

overlooking knowledge as its on object of study. That the framework falls under the umbrella of 'social realism' means that knowledge is considered both socially constructed and real, in that it exists 'outside' of the minds of either individual or the collective. As a sociologically-based framework, its goal is to find 'what lies beneath', or, as Maton puts it, what the 'rules of the game' are [27]. LCT is also a practical framework, where revealing or making explicit these characteristics has implications for practice [28]. For instance, in a study by Howard and Maton [29] on technology use in Australian high schools, they found a significant difference between teachers of different disciplines. Codifying teachers (and student) responses to survey questions revealed that this was due to their different implicit disciplinary identities. Making these characteristics explicit then allows better understanding of the use of technology in these classrooms. In terms of student understanding, Georgiou et al., [30] found that this was facilitated by identifying and characterising a particular element of knowledge, its abstraction. The research demonstrated that there was an underlying code in terms of abstraction when students answered typical exam-style questions: abstract, but not too abstract (e.g., the referring to the energy of phase change, rather than the employment of the ideal gas law). Due to its focus on knowledge and its practical utility, LCT was considered an appropriate framework to shed light on DSPs used as assessments in the tertiary context.

LCT consists of 'dimensions' which focus on a different element of practices known as its organizing principles. Three of these dimensions, Specialization, Semantics and Autonomy, have been significantly developed and applied in empirical research to address issues in education (and beyond). Specialization focuses on 'knowledge-knower structures' and is founded on the premise that 'practices are about or oriented towards something and by someone' [28] (p. 12). Thus, analytically, Specialization conceptualizes the relationships between practices and their object (known as epistemic relations) and practices and their subject (known as social relations). For example, Physics is a discipline known to be represented by stronger epistemic relations and weaker social relations: 'possession of specialized knowledge, principles or procedures concerning specific objects of study is emphasized as the basis of achievement, and the attributes of actors downplayed' [28] (p. 13). As a contrastive example, weaker epistemic relations and stronger social relations reflect cases where 'specialized knowledge and objects are downplayed and the attributes of actors are emphasized as measures of achievement, whether viewed as born (e.g., 'natural talent') cultivated (e.g., 'taste') or social (e.g., 'feminist standpoint theory')' [28] (p. 13). In Semantics, semantic structures are explored, whose organizing principles are determined by two constructs which vary in strength: semantic gravity and semantic density. Semantic gravity refers to the degree to which meaning relates to its context (the stronger the semantic gravity, the more strongly meaning relates to its context, the weaker the semantic gravity, the less strongly meaning relates to its context). To focus our analysis, we are employing the concept of 'semantic density' from LCT, because semantic density conceptualises complexity and the aim of our particular assessment products is to communicate complex ideas to a non-expert audience. Maton and Doran [31] elaborate on semantic density:

> *'Semantic density' . . . conceptualizes complexity in terms of the condensation of meanings within practices (symbols, concepts, expressions, gestures, actions, clothing, etc.). The strength of semantic density can vary along a continuum. The stronger the semantic density (SD+), the more meanings are condensed within practices; the weaker the semantic density (SD−), the fewer meanings are condensed. Put another way, semantic density explores the relationality of meanings: the more meanings are related, the stronger the semantic density. (p. 49).*

LCT studies suggest that negotiation of semantic density; strengthening (increasing complexity) and weakening (decreasing complexity or 'unpacking') is a key aspect of building knowledge in classroom practices [32–36]. In a study by Maton [36], for example, teacher presentations exhibit stronger and weaker moments of semantic density reflecting the 'unpacking' (describing elements of the concept of 'cilia' as little hairs that perform specific functions) and 'repacking' (a process whereby these functions are summarised with others in a comparative table) of scientific concepts. Theoretically,

this is described as the 'semantic wave' and has subsequently been identified as important in a range of different contexts.

Though diverse, studies in LCT have utilized some common methodological approaches. In terms of coding practices, a 'translation device' might be used. The translation device acts as a 'translation' of theoretical constructs (such as semantic density), to data. Though translation devices have not yet been developed for all constructs (nor are they necessary for each context) in LCT, a translation device *has* been developed for the analysis of language and thus, in this case, the translation device offered the principles by which coding of the semantic density of language occurs to represent different levels of complexity [31]. The main distinction occurs between 'everyday' and 'technical' language, with subdivision in each of these two main categories. The approach is an analysis of discourse, but not necessarily 'discourse analysis', since the analysis is sociological, focusing on characterizing the knowledge rather than the way it is expressed linguistically. Nevertheless, being an analysis of language, influences from Linguistics, specifically, Systemic Functional Linguistics, is apparent [31]. Systemic Functional Linguistics (SFL) is a theory of language commonly used in conjunction with LCT [37,38]. Some key ideas that originate from SFL and are relevant to this paper include 'technicality' and 'informational density', which were characteristics identified by SFL scholars as being key features of scientific discourse, as found in a range of 'expert' science texts, such as textbooks [39–42]. Technicality refers to the degree of specialization of meaning, where the crudest distinction occurs between 'everyday' meaning and meaning constructed within a particular field (e.g., of Science) [41,42]. Lexical density, according to Halliday and Matthiessen in SFL [42], refers to how much meaning is 'packed' within a text, and essentially 'counts' the number of 'content' words as a proportion of total words in a ranking clause. Theoretical studies have revealed certain profiles of language, such as the fact that spoken language is less lexically dense than written. Identifying these characterisations is important for learning, because, as Shanahan and Shanahan state [43], the complexities of language increase through schooling whilst explicit instruction in this area decreases. In science, it is generally understood that communication skills, particularly beyond the written form, are difficult to assess. In the assessment criteria of the task that is the subject of this paper, for example, though clearly stating that the text requires technical terms to be well-defined and pharmacological concepts 'conveyed to a general audience', interviews with the creators of the resources, as analysed in a separate study, give a sense that what is valued instead is 'getting the science right' [10]. The perception is that the 'communication' part is not explicitly taught or assessed:

> *if you are trying to teach students effective communication, we did not really get an assessment brief I did not really know what I was actually meant to be producing.*

> *If you are teaching these skills- in my experience they do try and teach communication skills but a lot of the time they do not mark towards the communication skills, they mark towards the content that is communicated. So even if it is communication poorly, you can still go okay.*

To consider assessment of digital products that include complex arrangements of representations and dynamic elements, like the DSPs discussed here, is substantially complex. To address this challenge, a range of different approaches is necessary, and as some have argued, this requires a new interdisciplinary field of research [44]. This work reflects the development of a preliminary framework, based on LCT, that aims to provide clarity around elements that have been difficult to assess in more complex assessment products, such as DSPs. The analysis is a first step for providing a literature base that supports the use of DSPs as assessments in higher education.

## 2. Materials and Methods

The theoretical approach was developed using a close analysis of two student-generated texts submitted as part of a third-year pharmacology subject at an Australian university. These texts were collected as part of a wider program of research supported by a national grant focused on learning

through student-generated digital media [45]. As part of this project, the full sample (*n* = 41) of digital media products, collected over a three-year period (2015–2017), were characterised in an iterative process involving the four researchers. This process produced a 'Variety-Quality' or 'VQ' Matrix, which acted as a way to easily glean some of the characterising features of the products. These products varied in many ways, as they were collected from a variety of universities, subjects and included a range of different types of assessments. For example, nutrition students constructed a digital product to suggest what a culturally appropriate diet should include for a resident in a retirement home and pre-service science teachers constructed a digital resource addressing a common secondary science misconceptions. This variety is captured in a separate paper [10]. The two products selected to develop the conceptual framework were from the same university subject, assessment and year but one achieved a higher mark. Table 1 depicts some characterisations drawn from the VQ Matrix for these two products. Excerpts from both are provided in the clip in the Supplementary Materials. Notably, they are both approximately five minute, standalone creations completed by students in a third-year pharmacology assessment at an Australian university and their purpose was to summarise a technical literature review and communicate complex information to a non-specialist audience (their peers in the subject, who are assumed to have general scientific knowledge only). In addition to these products, course documents (subject outline and assessment rubric) were collected and interviews were conducted with the subject coordinator and creator of the resources. Excerpts from these additional documents will be used for illustrative purposes only. Reporting of these analyses can be found in [46,47].

**Table 1.** Characterisation (Variety-Quality (VQ)) matrix for the two sample texts.

| Text | Details | Assessment | Audience | Basic Form | Accuracy |
|---|---|---|---|---|---|
| Malaria | Length: 4:53 Subject: Principles of Pharmacology Year: 2015 | Topic: Is Ferroquine an ingenious anti-malarial? Mark: 5/5 | Non-specialist | 95% image-narration-text, 5% animation explanation, informational | High |
| Multiple Sclerosis (MS) | Length: 4:54 Subject: Principles of Pharmacology Year: 2015 | Topic: What are the latest developments in therapeutics for the treatment of multiple sclerosis? Mark: 4/5 | Non-specialist | 100% slowmation explanation, informational | High |

Various software programs were used to analyse the data. Given the still-developing field of multimodal research, no one product was able to perform all necessary analytical functions. The transcript, representing the verbal narration, was structured by clause and individual frames (or a group of individual frames) were then matched to each clause. This structure, including time stamps, was analysed in Excel for the semantic density analysis for narration. These data were ultimately imported into NVivo for further analysis, specifically for the semantic density analysis for the images, which occurred at the frame level. These analyses were returned to Excel, to consolidate and manipulate the data for the quantitative analysis. The narration and images were considered separately and then subsequently integrated in an additional analytic process.

In terms of the coding, because the analysis was based on LCT, this might be considered 'deductive coding', where the key stages include 'developing the code manual' and 'testing the reliability of codes [48]. Each part of the analysis was conducted in consultation with the relevant experts (SFL, Science or both) and involved initial coding, negotiation and final allocations. After negotiations, final codes were established at or near 100% agreement.

All subjects gave their informed consent for inclusion before they participated in the study. The study was conducted in accordance with the Declaration of Helsinki, and the protocol was approved in May 2016 by the Human Research Ethics Committee of the University of Wollongong (protocol number HE16/165).

## 3. Results

Analysis and results are presented in two main parts. Analysis of the two sample texts begins with semantic density analysis of the language in the narration or audio component of the texts, which is followed by analysis of the images or visual component. In the final part of the results section, we offer a possible method for combining these analyses to make judgments about the texts more holistically. While we note the need for caution in conducting these analyses separately, particularly in terms of what could be lost when the components are treated in isolation, we illustrate the processes of our analyses here to provide detail on the analytical approach. Because the texts represent a new form of assessment in tertiary science subjects and our aim is to illustrate how the students have negotiated the complexity of the scientific information across multiple modes in a digital product, the characterisation is a first step and its utility will be discussed, along with its limitations, in the discussion.

### 3.1. Language (Narration) Analysis

In this section, the analysis of the language (narration) portion of the text is presented. This involves a presentation of the coding and quantification of complexity using the concept of semantic density and with reference to the use of the translation device discussed in the Introduction section.

To analyse the language portion of the texts (the narration as audio track), the LCT concept of 'semantic density' was employed. Semantic density refers to the 'condensation of meaning' and provides an indicator of how complexity is built and represented throughout a text [27]. A coding structure developed by Maton and Doran [31] was implemented. This structure, known as a 'translation device', considers various levels of meaning –from the individual word, to the sequencing of paragraphs. The device looks at both discrete 'degrees' of semantic density (density—the 'amount' of condensation of meaning) and how it is built throughout the text (condensation). Maton and Doran's [31] paper identifies that coding can occur at various 'units' (word, word group, whole text) and to various degrees of delicacy (type, subtype, sub-subtype). In this analysis, the semantic density was applied at the word level only. The analysis occurred over two cycles, with the researcher consulting specialists of SFL and LCT to ensure the tool was used consistently.

The analysis employed the same annotation as Maton and Doran [31], depicted in Table 2. For example, annotations included font changes, capitalisations or bold letters to indicate type and/or subtype in the wording tool.

**Table 2.** Annotation and coding for wording type and subtype.

| SD | Type | Subtype | Examples |
|---|---|---|---|
| +　↑　↓　− | Technical (meanings are given by their location within a specialized domain of social practice) | CONGLOMERATE (comprise multiple distinct parts that each possess a technical meaning) | Monosaccharides Ferroquine Leukocytes |
| | | Compact (comprise a single part with a technical meaning) | Force Parasite |
| | Everyday (meanings are given not by their location in specialized domains but rather through their usage in commonplace practices and contexts) | CONSOLIDATED (encode happenings or qualities as things) 'Happenings' are processes or events that are normally realized by verbs and 'things' are elements or items (physical or intangible) that are normally realized by nouns | Death Prevention Action |
| | | Common (leave happenings or qualities as qualities) | Moves Mosquito |

The excerpt below shows the annotation and coding of one clause in the transcript for the narration for the Malaria text.

This is attributed to the inability of the transporter producing Chloroquine resistance to remove Ferroquine from a digestive vacuole.

Given the current research is making a judgement about how complexity is negotiated in the text, we quantified degrees of complexity as part of the analysis. In previous LCT studies ordinal or interval measurements have been used to quantitatively determine 'strengths'. For example, Georgiou et al. [30] assume ordinal data when identifying three different semantic gravity strengths, because one simply represents a stronger or weaker 'degree' of context dependence. In the current study, quantitative assignments were given to the different word types, reflecting different strengths of semantic density. The scale was (approximately) interval, with the assumption that there was a larger gap between the technical and everyday types than between the subtypes in each category. This assumption is already qualitatively supported by work in the field of linguistics (SFL), although work is continuing [41,42]. The quantitative assignments are, therefore, meaningful when considering degrees of complexity as well as relative differences when considering technical and everyday distinctions, but absolute differences are not meaningful. The assumption of interval data allows us to perform the arithmetic operations outlined in the following section.

The degree of semantic density (how technical the text is at this point) was measured relative to the clause. The clause is considered the unit onto which different kinds of meanings are mapped, according to Halliday and Matthiessen in SFL [42]. The quantitative measurement, therefore, reflects relative strengths of semantic density of the word per clause, across the text. Everyday common words attracted zero value, everyday consolidated attracted 1, Technical compact, 3, and Technical conglomerate 4. If multiple counts of the word subtype were present, they were added together. For example, in the Malaria text presented below Table 2, the clause contains one everyday consolidated word type (inability), three technical compacts (resistance, digestive and vacuole) and two technical conglomerates (Ferroquine and Chloriquine). This would attract a quantitative assignment of 18. In the first instance, the overall average degree of semantic density for both sample texts was calculated by adding the value for epistemic semantic density for each clause and diving by the total number of clauses (Equation (1). $SD_{AVE}$ = 4 for the Malaria text and $SD_{AVE}$ = 8.5 for the MS text. This modest measure simply indicates that on average, MS had relatively stronger semantic density than Malaria.

$$SD_{AVE} = \frac{Sum\ SD\ values}{total\ no.\ of\ clauses} \tag{1}$$

The quantitative assignments for the word subtypes were then displayed as a function of clause number to reflect the relative strength of semantic density for the narration across the text for both Malaria and MS, as shown in Figure 1.

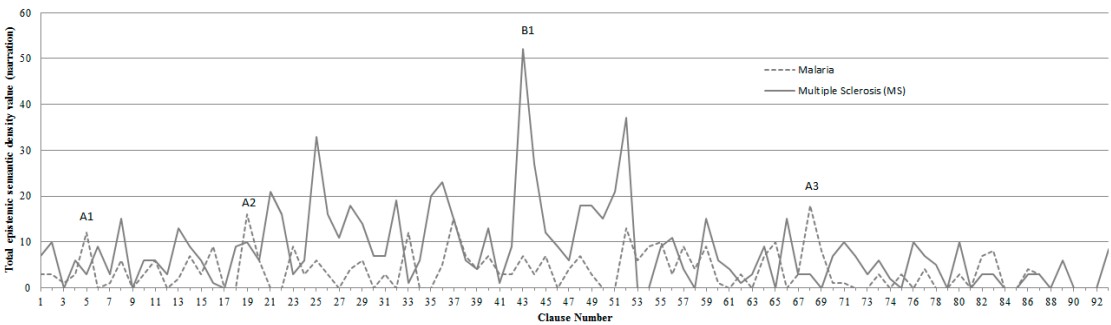

**Figure 1.** Relative strengths of semantic density per clause for the narration in Malaria and MS across the text. Identified points (A1, A2, A3, B1) are discussed later in the paper.

Figure 1 illustrates that Malaria exhibited a consistently 'waving' profile that had, on average, lower semantic density, whilst MS had a higher semantic density on average, working up to a peak near the middle.

### 3.2. Image Analysis

Semantic density was also used to consider the images in these two sample texts. In this section, an analysis of the images, as separate from the narration, was conducted. The analysis aimed to categorize the different types of images used in the texts.

Borrowing from Maton and Doran's [31] coding structure for text, a distinction was made between 'technical' and 'everyday', with a further distinction within each category. These categories were created iteratively to achieve a level of consistency between two coders (with expertise in LCT and physics and expertise in Science, respectively). The four levels were consistently identified and did not introduce any significant disputes, although these texts happen to be the most straightforward types, containing only image combined with narration and the categories broad. The categories are provided in Table 3.

**Table 3.** Image categorisation with semantic density (SD).

| SD | Image Type |
|---|---|
| + | **Scientific complex** |
|  | (i)  Images that represent a scientific object(s) or representation in a traditional format. |
|  | (ii)  Images that represent a scientific process. |
|  | **Scientific simple** |
|  | Images that represent a scientific object or representation in a simplified or 'non-traditional' format. |
|  | **Everyday real** |
|  | Single/isolated images that are photographs (or 'photoreal') of real objects |
|  | **Everyday illustrative** |
|  | Single/isolated images that are illustrative or stylistically (but not 'technical') representative of the 'natural' version (e.g., cartoon-like images) |
| − | **Text image** |
|  | Images that include text and symbols (including punctuation and not assigned a semantic density value). |
|  | Blank (blank screen) |

The most diverse categories were the two technical levels; scientific complex and scientific simple. Figure 2 displays examples of these categories from both sample texts. The example chosen from the scientific complex category (relatively strong semantic density), which was taken from the MS text, belongs to this category because it contains both a highly technical representation (the neuron in the 'brain') and many parts. The example chosen from the scientific simple (relatively weak semantic density), taken from the Malaria text, is categorised as such because it is a simplification of a technical representation (the parasite's digestive vacuole—which is what this image represents—is not shown in full or labelled, only the relevant features are presented).

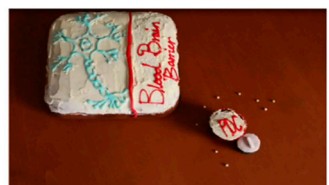
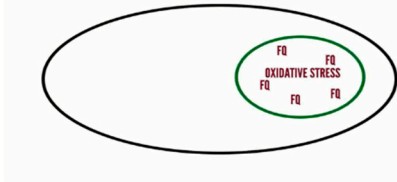

**Figure 2.** Example image semantic density categories: scientific complex image from MS (**left**), scientific simple image from Malaria (**right**).

The incidence of each category of image for each sample is presented in Figure 3 where duration is displayed as a proportion of total for each sample. The displays in Figure 3 illustrate key differences in the visual component of the texts in the current study.

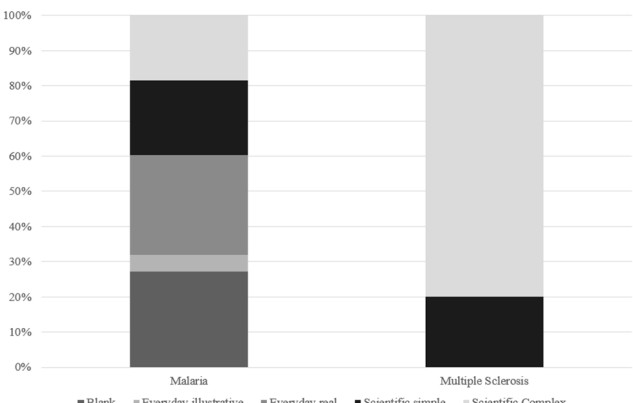

**Figure 3.** Duration (as a proportion of total) of each type of image in Malaria and MS. Relative strengths of semantic density increase down the column.

The key differences in the use of images for the two products were types used and relative durations. Malaria used a greater variety of image type, including a blank screen while MS used more of the 'Scientific Complex' type, easily the most prevalent choice in this text (accounting for 80% of all images used). Most of these different categories did not occur contemporaneously, that is, there were very few moments where two different categories were present in a frame at the same time, thus, Figure 3 represents these categories as reflecting the composition of the text. However, in terms of 'text image', this often appeared overlaid with other images and was thus excluded from Figure 3 for clarity. 'Text image' was used in Malaria for a total of 1 min and 20 s and in MS for a total of 1 min and 2 s.

*3.3. Image and Narration Combined Analysis (Points of Interest)*

In this section, more detailed analysis of 'points of interest' in the text is presented. This involves looking at the text as a whole at these points (looking at all semiotic resources used at these points to communicate complex scientific concepts). Each point will be presented in turn.

Since the focus in this paper is on how the creators of these digital explanations negotiate the complexity of meaning through the text, points of interest, points of 'maximum' density, were identified (A1, A2, A3 and B1 from Figure 1). These points of interest were subsequently more closely examined to consider how the complex ideas were negotiated across both the narration and visual sections of the text. Maton and Doran [31] refer to this as 'condensation' and this is determined (in language) by considering how semantic density is strengthened through the 'clausing' and 'sequencing'. Since we are interested in how meaning is built across multiple resources in the texts, we draw on the notion of condensation to describe what is happening at these key points in the student-generated texts more generally.

The display in Figure 1 includes an average density calculation for each clause of the blended media. The points labelled A1, A2, A3 and B1 were chosen because they are high points indicating the presence of increased complexity. For the Malaria text, three different points (A1—clause 5, A2—clause 19, A3—clause 68) were chosen from maxima in the combined text-image 'quantities. The Malaria text was more varied than B, so three points were necessary to represent the different 'points of interest'. For MS, only a single maximum was found (B1—clause 43), as the patterns of negotiation were similar throughout the text. Though the quantification of image is not shown here for ease of readability, each of these points were a maximum in terms of combined language and image values. It should be noted that the three highest points on the MS text were each higher than any of the points from the Malaria text. Each point will be expanded on below.

### 3.3.1. Point A1 Malaria

At this point in the text (A1, which is the first point of interest in the Malaria text), the narration exhibits a relatively high value for the strength of semantic density. Four technical compact terms are used in the one clause, with three consecutive terms amassed at the end (Figure 4).

**Image:**

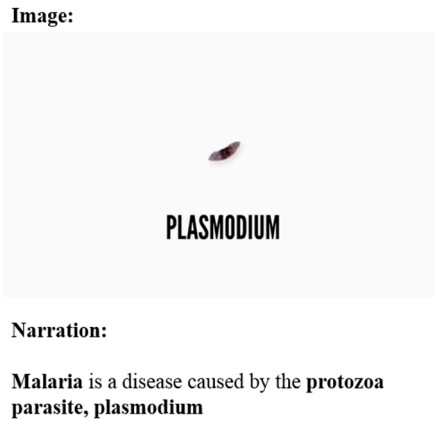

**PLASMODIUM**

**Narration:**

**Malaria** is a disease caused by the **protozoa parasite, plasmodium**

**Figure 4.** Clause and image (s) present at Point A1 (Malaria text).

In order to negotiate this complexity, two distinct techniques are apparent. The first technique involves the weakening the semantic density in the interplay of the image and narration on screen at the same time. In terms of the relationship between the image and the narration, we see that the image which accompanies the technical narration 'protozoan parasite, plasmodium' is in fact an 'everyday real' image of a plasmodium (Figure 4). The image is also accompanied by the text on screen 'plasmodium'. The complexity in the narration is, therefore, reduced or 'unpacked' by the signalling done by the image and with the use of this label. The second technique involves sequencing. In the clause following Point A1: 'and can cause fever, headaches, chills and vomiting in a patient', we observe no technical or everyday consolidated terms. This represents a decrease in the degree of complexity; semantic density has changed from a relatively stronger to a relatively weaker value in the space of time of these two clauses. This is also true for the preceding clause (see Figure 1). Together, these two techniques act to signal that the plasmodium is a 'thing' and this 'thing' can cause a range of familiar symptoms. This technique is used repeatedly in this text and is demonstrated again in Point A2.

### 3.3.2. Point A2 Malaria

Point A2 represents the second highest peak in terms of the strength of semantic density for Malaria and is another illustration of the negotiation of complexity we explored at point A1. In terms of the image and narration shown in A2, we can see that three everyday real images of 'drugs' are presented (Figure 5), supporting the considerably technical terms used in the accompanying narration. Like the signalling of the 'plasmodium', here, the images of drugs are used as a placeholder for 'mefloquine, doxycycline and chloroquine'.

**Images:**

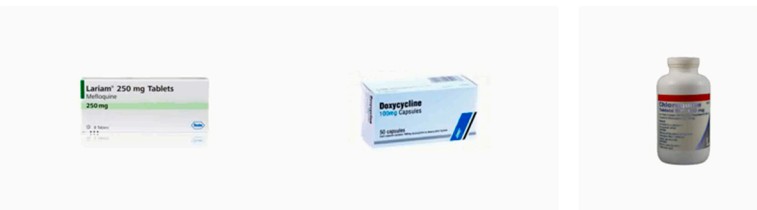

**Narration:**

Current **Antimalarials** include **MEFLOQUINE, DOXYCYCLINE** and **CHLOROQUINE**

**Figure 5.** Clause and image (s) present at Point A2 (Malaria text).

In the clause immediately following this one: 'Chloroquine is the most widely used amongst developing nations due to its effectiveness and low cost of production', we see that two everyday consolidated ('effectiveness' and 'production') and one technical conglomerate ('Chloroquine') words are used. In the two clauses that follow: 'If we've got so many drugs already' and 'why do we need another?' no semantically dense terms are used. Again, this is also true for the preceding clause. Thus the two techniques used to negotiate complexity in Point A1 are also used in Point A2.

### 3.3.3. Point A3 Malaria

The peak at point A3, exhibits the highest value for semantic density overall. It was also the point of highest value for semantic density in language per clause for the Malaria text. We see that this image is also semantically dense (at the second highest level according to the image categorisation in Table 3). This image is coded as 'simple scientific', not because it is 'simple' but because aspects of the scientific diagram have been simplified to only display the process described in the narration.

At this point in the text, we do not see the tandem use of the two techniques outlined in Points A1 and A2. Instead, though the image used is not the most technical or complex, both image and language (narration) together make this point the most technical in the Malaria text (Figure 6). This point provides an example of 'building' and is extremely important, as it fundamentally answers the question posed for the assessment (Is Ferroquine an ingenious anti-malarial?). The creator concludes that Ferroquine *is* ingenious because it is not as prone to resistance as compared to its main alternative, Chloroquine.

**Image:**

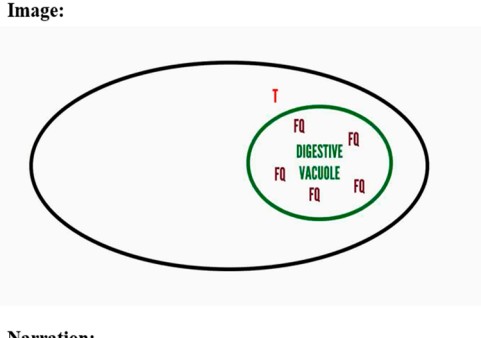

**Narration:**

This is attributed to the INABILITY of the **transporter** producing **CHLOROQUINE** resistance to remove **FERROQUINE** from a **digestive vacuole**

**Figure 6.** Clause and image (s) present at Point A3 (Malaria text).

The key mechanism behind this statement was outlined in an earlier section of the resource: Ferroquine accumulating in the digestive vacuole and causing oxidative stress, killing the plasmodium.

The building of the image from Point B3 was slowly created throughout the text, with sections added and the same diagram used multiple times (Figure 7), to show the different stages of the process. Simple scientific images were used, and these act to highlight the important parts of the mechanism (rather than using the more 'textbook' style representations of this microbiological process). The image and narration work together to develop the explanation, building on themselves (across the text) and each other. This is the single example of such building in this text.

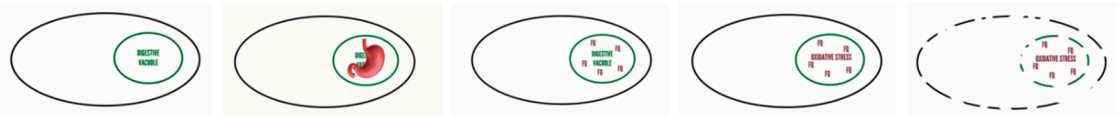

**Figure 7.** Images as part of a seven-clause sequence that builds up diagram of plasmodium processes (Malaria text).

### 3.3.4. Point B1 MS

In this excerpt, from MS: Multiple Sclerosis, the semantic density in the narration was well above any clause from the Malaria text. There are 15 technical compact terms and two everyday consolidated terms. Looking to the images used to support this narration, these are also coded the highest level of semantic density, both because they contain 'standard' scientific diagram (the neuron—as would be represented in a textbook, for example) and multiple technical parts (depictions of immune cells, drugs and a structural depiction of the blood bran barrier) (Figure 8).

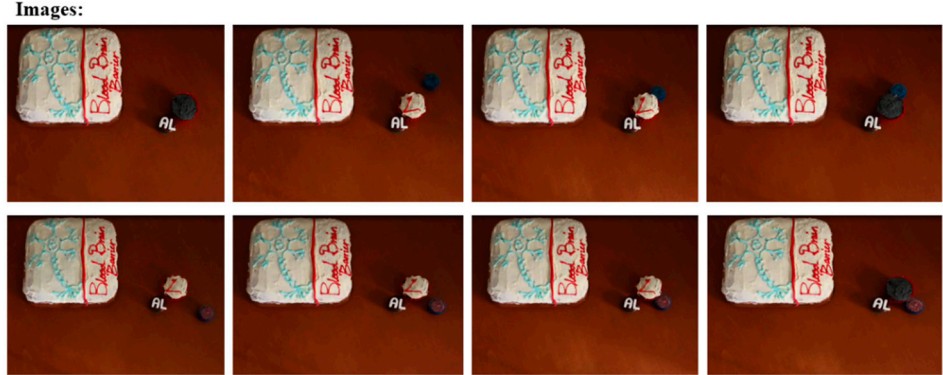

**Figure 8.** Clause and image (s) present at Point B1 (MS text).

In this example, we do not see any of the negotiation techniques used by the creator of Malaria. The images do not provide a 'placeholder' for the technicality present in the narration and in fact, are extremely technical and in need of translation themselves (it is not entirely clear through either labelling or representational form/symbol what each of the parts represent). This section can be described as building and the images are in fact doing most of the building work; they demonstrate the process of cell death described in the narration. Two specific forms of cell death (apoptosis) are represented (represented by the first three and last two sample images in Point B1 box). These images (used in a 'stop motion' approach) communicate the message that the drugs are used contribute to cell death, which is important as the accumulation of white blood cells in the brain (left of the blood-brain barrier) causes the damage to the myelin sheath (responsible for symptoms of MS).

In considering how this building is placed in the context of the whole text, we notice that rather than this being the only case, as with Malaria, there are multiple distinct sections just like this one,

making the semantic density much stronger overall. For instance, the clause that follows this one is a segment that is similarly relatively semantically dense but unrelated to the clause at point B1: 'Daclizumab binds to the Tac epitope on the IL-2 receptor on the CD25 chain on T cells'. In fact, this text is simply a series of distinct, very technical explanations just like these ones, with no negotiation or break from the technicality. In terms of the building, though the building is necessary, the accelerated growth of complexity within the images coupled with the lack of negotiation within the narration, makes this attempt at building knowledge for the purpose of communication less successful than those in the Malaria text.

## 4. Discussion

Analytical approaches in the fields of technology and science education in terms of representing knowledge are still in their infancy and, as such, support for students and teachers in relation to developing and assessing effective multimodal communication in digital products is not well developed. This is important because this area, working with multiple representations facilitated by technology, has potential to be holistically beneficial to a student in terms of their science education. In this paper, we developed a preliminary conceptual framework that draws on LCT in order to understand how to assess student understanding of science and communication through digital products. In this section, therefore, we discuss both what this type of analysis can tell us about the practical issue of DSPs as assessments and what the affordances and limitations are of this particular theoretical approach in terms of research in multimodality and technology use.

### 4.1. Multimodal Digital Assessments

The DSPs studied in this project involve the use of technology to allow knowledge to be represented dynamically through image, text and narration. As mentioned in the background section of this paper, the potential for digital products as authentic and holistic assessment is being increasingly recognised. In the words of a student undertaking this task: 'this sort of task . . . I learn better by researching it and figuring it out myself than just being taught it because I have to get it, because if I don't, I don't get the marks' [10].

However, as discussed in the Introduction, the assessment of communication in science is often sidelined due to a focus on 'content' and there are limited frameworks in the literature which can be utilised. In this paper, we focus on one element, understanding the nature of communicating complex scientific concepts. In this space, the LCT concept of semantic density is employed, in order to more reliably assess the level of understanding represented in the products, of both the science and communication (how complex and how much 'condensation' is appropriate for each audience). Analysing the digital products in terms of the relative semantic density expressed in both the image and the narration first revealed the two texts to have different overall 'quantities' of average semantic density. The considerable difference in the levels of semantic density in the two texts (as well as across the whole sample) demonstrates that judgement around the level of semantic density, or technicality, used in a text for a specific audience is inconsistent. These quantities could be used as a crude measure of the level of technicality for the students, calling on them to consider whether this was appropriate or not, for example when compared to other texts for the same audience or a model text provided by the instructor.

More importantly, two distinct techniques that 'control' complexity in some way were apparent in both these texts: negotiation and building. They provide us with insight into how complexity is communicated effectively. In the Malaria text, these two techniques were used to great effect, as signalled by the full marks awarded for the assessment (Table 1). One negotiation technique involved 'place-holding': using common-sense language or images at points where technical terms or complex processes were introduced. When listing the names of the various drugs, for example, a picture of a pill box was shown. This idea is supported by comments made by the creator of Malaria in response to a question about how they attended to audience:

> *So a lot of the decisions I made was just because if you are an unspecialised audience you don't really care about the science behind it . . . so it's not a detailed image, there's not information image: it's just more this is a mosquito, that's a mosquito, so I'm saying mosquito here's a picture of a mosquito.*

This negotiation attempt acknowledges that the text is at a relatively semantically dense section (in the section the creator is referring to, the term 'female anopheles mosquito' is used) and negotiates this relatively strong semantic density by simplifying the message in the image (mosquito). In the Malaria text, there are many of these attempts at negotiation; the use of a 'placeholder' allows easily accessible meaning to be condensed into technical terms. The second technique involves the employment of less technical sections preceding and following sections with relatively stronger semantic density, as outlined in the previous section and in the peaks in Figure 1. This could reflect one aspect of the required 'balance of detail' across the text, as specified in the rubric.

However, at Point A3 we see instead a different technique, building. At this point, there is no negotiation or common-sense meaning represented in the image and both image and narration are relatively technical. In this section, the sustained technicality is necessary because a complex idea needed to be communicated and this required careful building across image and language and throughout the text. However, this only happened once in the text and, as such, points of negotiation were intentional acts to climax to the 'building' activity that communicated the central message of the text.

In MS, we see evidence of multiple points of building but not negotiation of complexity. Point B1, for example, builds complexity in reflecting the process described in the narration in the imagery. However, whole process is described in only two clauses (this includes the clause in Point B1 and the previous clause only). The clause following point B1 represents a completely different process. In fact, this whole process is described in less than 15 s, only to be repeated immediately after. What is significant about this is that the text essentially consists of a series of such explanatory sections; very complex 'building' processes that are neither linked to each other or negotiated by the narration or image. The narration stays extremely semantically dense; it is not clear that there are any attempts to, as the assessment criteria state: define 'Technical terms', use 'language appropriate for a general audience' and ensure 'Material is relevant and an appropriate amount is provided (balance of detail vs. general overview)'. This resource also does not make use of labels. This is ultimately a less successful attempt at communicating an idea to a non-specialised audience, though the complexity is there, the level of technicality (semantic density) is misjudged and the building is too abundant and too rapid to have the same effect as Malaria.

Semantic density could potentially offer a language with which to communicate how complexity manifests across multimodal forms in science. This could be useful in assessment in order to provide instructors with more specificity around communicative elements of a task, as well as provide pedagogical resources to assist students in identifying technicality and complexity, and how to negotiate them. For instance, the quantification of complexity could act as a quick check of the level of complexity of texts created as communication objects for particular audiences. Different 'complexity' ranges could be identified as more or less appropriate, flagging to the maker that the text should be further developed. The two techniques identified here could be included as part of a 'toolkit' of techniques that can be provided to students to help them decide how to consider the audience when producing communicative texts. More generally, this knowledge helps makes these elements of knowledge more explicit. That is, the 'rules of the game' are made more visible to students, providing them access to otherwise hidden disciplinary ways of knowing and representing.

### 4.2. Use of a New Approach to Analyse Digital Products

The second matter we will discuss involves considering the merits of this preliminary conceptual framework to analyse digital products. The previous section demonstrated its utility in the practical sense of assessing communication skills, and this section, therefore, focuses on the theoretical and methodological implications, benefits and limitations of LCT.

Theoretically, LCT is a 'socially grounded' approach focused on 'decoding' knowledge. In this particular case, we were interested in one aspect of these texts: how meaning or complexity is built. Semantic density was, therefore, used as a construct with which to understand this aspect. In terms of the methodology, there are two key strengths that are important to highlight. First, the 'translation device' offers a way of limiting subjectivity and increasing transparency in methods involving coding. The coding activity was very consistent, and few discrepancies were found, avoiding an otherwise problematic issue widely known to affect qualitative methods. The second main methodological advantage was the nature of assigning 'values' of semantic density. Assigning a value to data affords the advantage of further analysis and of more straightforward communication of key ideas. In this case, the 'degree' of complexity was quantified and allowed a clear representation of the degree of average semantic density in the text, as well as how it changed throughout the text. The sliding scale present in LCT across a range of concepts also allows for infinite gradations, offering the possibility of relatively simple or significantly sophisticated analysis. Further, since we are simply discussing 'relative' values of semantic density, we are comparing data points to each other, rather than to an absolute. These advantages are becoming increasingly recognized in research where knowledge is key [49].

### 4.3. Limitations and Further Research

As a 'first step' in the analysis of a complex object, both the practical and theoretical elements of this study exhibited limitations. In terms of the practical aim, to clarify characterizations of how complexity is communicated in a student-generated digital product, known as a DSP, though detailed, the analysis was based on only two samples. Thus, the principles that were identified are likely only a subset of a larger set of principles.

Theoretically, dealing with meaning making across modes was a challenge. Often, it was difficult to know how to 'treat' a resource (or indeed how to label them). In the MS text, for example, the 'images' were actually placed consecutively to form a 'slow animation' or 'slowmation' [12]. In treating these stills as images, it is possible that we might omit consideration of factors associated with how ideas progress over time. In terms of coding of the images, though our two samples proved relatively straightforward, our categories were also quite broad. Particularly in terms of the 'simple scientific' and 'complex scientific' categories, the two relatively strong semantic density categories. Determinations of complexity across different disciplines (e.g., a chemical representation of a molecule or a display of a graph), could be a challenge. Furthermore, the use of language as image (text on screen) would be similarly difficult to manage, as would any assumption about the 'primary' of the resource used (e.g., narration as being the 'main carrier' of meaning). The quantification of semantic density could also be open to critique; such quantification isn't common so how to treat it, quantitatively and statistically, is still to be established.

Further research could continue this analysis with a larger number of texts with more complex combinations of modes (including animation), to confirm the existence of these techniques for successfully communicating complex scientific concepts in dynamic media, as well as identify any others. Other elements of knowledge, including the degree of abstraction or the making of 'interpersonal' meaning could also be explored in order to more explicitly capture and, therefore, more precisely assess these aspects of knowledge, where appropriate. Increasing our understanding of more 'open' assessments such as these DSPs is important if we want to encourage a more 'holistic' teaching and learning experience for university students.

### 5. Conclusions

This paper represents a detailed analysis that aimed to conceptualise complexity in student-generated digital products. The approach had both a practical and theoretical purpose. Practically, it presented a new analytical approach to understand how communication in digital products can be captured, based on Legitimation Code Theory (LCT). The analysis used the concept of semantic density from LCT and was able to provide a quantification of 'complexity' and identify

particular techniques used in the Digital Standalone Products to manage it. Identifying these techniques allows us to more clearly discuss what is happening, specifically, how complexity is being communicated (successfully or otherwise), across the multiple modes of a digital product. Theoretically, the detailing of how LCT was used, including its benefits and limitations, is a contribution to the field of multimodality and a first step in being able to describe how meaning is made across multimodal texts. The complexity of dealing with multimodality requires significant theoretical and analytical developments in order to fully take advantage of the potential offered by new technologies to support student learning.

**Supplementary Materials:** The following is available online at http://www.mdpi.com/2227-7102/10/1/18/s1. Clip: Malaria and MS excerpts.

**Funding:** This research was funded by the Australian Government Discovery Project Grant Scheme under Grant number DP160102926.

**Acknowledgments:** I wish to acknowledge Yaegan Doran, Pauline Jones and Annette Turney for contributing to the analysis. Chris Hyland and students are recognized as participants. Wendy Nielsen, the chief investigator of the project, coordinated this aspect of the data collection and contributed to the summaries in the VQ Matrix.

**Conflicts of Interest:** The authors declare no conflict of interest. The funders had no role in the design of the study; in the collection, analyses, or interpretation of data; in the writing of the manuscript, or in the decision to publish the results.

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
