# Peer review of "Characterising Communication of Scientific Concepts in Student-Generated Digital Products"

_education, doi:10.3390/educsci10010018_

Round 1
Reviewer 1 Report
The paper is written according to the instructions for authors.
Tables and figures are in correlation with the text in the paper.
Methodology described in the paper is very clear.
The results are very clear and systematically presented.
Paper is classified as the original scientific paper.
Author Response
Thank you for your feedback (no changes required).
Reviewer 2 Report
Thank you for allowing me to review "Characterising communication of scientific concepts in student generated digital products."
Overall, it is a well written and interesting manuscript. I recommend some changes in order to make the material more accessible to a wider audience.
Define elements such as "tertiary science course" Consider including references to discourse analysis Clarify that the study centers on college-level material. The mention of high school on page 2 was slightly confusing. Eliminate words such as "hope" (e.g., pg. 2). Explain the five dimensions from pg. 3. Although not the focus of the manuscript, it provides context for the work. Make clear statements at the beginning of each section to explain what is contained in that section. Sometimes as a reader I wondered why I was reading details, only to find out at the end of a section. This is especially true of the results section. Explain being "drawn from a variety of university courses and included a range of different types of assessments" (pg. 3). Move the participant pool up in the explanation (pg. 3) "...third year pharmacology assessment..." Check to make sure that all citations outside of the literature review are contained within the literature review. Clarify if this is a theoretical or empirical work. It seems as if clear research questions are needed to lead to the study. Overall, the tables and figures are helpful. Can Figure 3 be reworked so that the material presented is accessible? Consider giving a clearly laid out example in a table to showcase some findings and the overall results. In places, this work reads as an evaluation instead of a research study, and this relates back to #9 (clarifying theoretical or empirical work). Could another case be added? If a "high" and "low" case are presented, it would be beneficial to see how an "average" case compares. Clarify the impact of this work. How does it potentially move the field forward? What gap in the literature does it fill? Clarify the limitations of the work.Author Response
Thank you for your comprehensive comments on this paper. All points are addressed in the table below.
|
Define elements such as "tertiary science course |
‘course’ was changed to ‘subject’ and ‘tertiary’ to ‘university’. References 1,2,10 and 11 include specifics examples of the range of subjects |
|
Consider including references to discourse analysis |
Discussion of SFL and analysis of discourse added to the final paragraph of the introduction |
|
Clarify that the study centers on college-level material |
Clarified in ‘materials and methods’ |
|
The mention of high school on page 2 was slightly confusing |
Added a qualifying clause justifying the discussion of research from primary/secondary contexts: ‘at all levels of education’, which is relevant in order to demonstrate the consistency of the research findings |
|
Eliminate words such as "hope" (e.g., pg. 2). |
These were removed throughout the paper |
|
Explain the five dimensions from pg. 3 |
This section has been reworked to discuss the dimensions that have currently been developed/developing and used to address issues in education research. Other feedback on this paper indicates that explanation of theoretical concepts not utilized in the paper might be extraneous, however, because this paper is so centrally about LCT, I have included a discussion of the two most used dimensions, Specialization and Semantics. |
|
Make clear statements at the beginning of each section to explain what is contained in that section (esp in results). |
Added |
|
Explain being "drawn from a variety of university courses and included a range of different types of assessments" (pg. 3). |
Two examples are included to clarify |
|
Move the participant pool up in the explanation (pg. 3) "...third year pharmacology assessment..." |
Added |
|
Check to make sure that all citations outside of the literature review are contained within the literature review. |
Done. Some of the sections in the results were moved up to the introduction (including a section on SFL/linguistics) |
|
Clarify if this is a theoretical or empirical work (also, could another case be added?) |
It is theoretical work that draws on empirical data but that nonetheless has implications for practice. This has been clarified in the manuscript (e.g., abstract, final paragraph in introduction, conclusion) and limitations with respect to sample size have been acknowledged. |
|
It seems as if clear research questions are needed to lead to the study |
Because this work is concerned with developing a theoretical basis for understanding these products, it doesn’t seem that research questions are a good fit. However, additional clarification has been added (see above) |
|
Can Figure 3 be reworked so that the material presented is accessible? |
I’m happy to amend this diagram but am unsure of which aspects are not accessible. |
|
Consider giving a clearly laid out example in a table to showcase some findings and the overall results. |
I attempted this but it seemed to add to the ‘dense-ness’ of the results and discussions section, which was identified as problematics by another reviewer. Thus, I ultimately made the judgement call that it wasn’t appropriate. |
|
Clarify the impact of this work. How does it potentially move the field forward? What gap in the literature does it fill? Clarify the limitations of the work. |
Clarification/elaboration has been added to a new limitations sections. |
Reviewer 3 Report
Review of “Characterising Communication of Scientific Concepts in Student Generated Digital Products”
17 Dec 2019
This article proposes a new way to examine student-created digital assessments. The paper is interesting, and I have a few questions to improve the clarity for those not familiar with LCT.
I notice that the references are across ages and school years/types. I think this is reasonable since the proposed method could be applied across multiple ages and years.
Is there an additional reference to back up the statement in line 61 with reference 24?
Lines 75-81: This is kind of link the difference between content knowledge, pedagogical knowledge, and pedagogical content knowledge. If these terms aren’t US-specific, they may be useful as an analogy.
Lines 98-102: Can you provide an example of “abstract, but not too abstract”?
Lines 110-115: Good. The rest of the paper focuses on semantic density. On the first read, I wanted an example of semantic gravity. But on a second read, I think you’d be OK without it. But if another reviewer would like it, go for it.
Section 2/Lines 125-141: Can you please clarify that these are undergraduate/graduate/both? Without seeing the previous paper, it is useful to know a bit about the demographics of the students providing these products.
Line 185: Good
Around Table 2: I wondered how you defined a “clause”. Is this just the standard language definition of a clause? Or did you use a different definition? Clarification on this would help the reader.
Lines 237-238: Please clarify if it is LCT & physics, or physics & science
Table 2 shows the formatting of the code (bold/caps/etc.) which is very helpful.
Figure 3: so this means there was no simple text image? Please clarify that I am reading this right.
Figure 1 is initially described as the semantic density per clause: the textual analysis. Then in line 288 it suggests that Figure 1 is for the blended media (text and image). Do you need two figures?
In section 4.2, lines 499-517 feels too detailed. Maybe put some of this in the introduction, or simplify it for the reader who is a novice to LCT?
The discussion could use a little more discussion of how teachers could use this. It currently has some, but at the more theoretical level. It would be nice to see a clear “we plan to make a tool for marking digital products” or some other statement to give teachers a reason to keep up with this research.
Author Response
Thank you for your feedback on this paper. I found the suggestions most valuable and hope that the response (below) address the identified issues.
|
Is there an additional reference to back up the statement in line 61 with reference 24? |
(Kumar et al., 2002) has been added
|
|
Lines 75-81: This is kind of link the difference between content knowledge, pedagogical knowledge, and pedagogical content knowledge. If these terms aren’t US-specific, they may be useful as an analogy. |
Whilst PCK is a framework that does have some overlapping considerations with LCT, It would be best to avoid introducing other theoretical frameworks that don’t explicitly relate to the analysis presented, in an effort to address the ‘complexity’ issues identified by the other reviewers, particularly as PCK literature relates more to school teaching and preservice teacher education, rather than the communication of complex ideas to a non-specialised audience. |
|
Lines 98-102: Can you provide an example of “abstract, but not too abstract”? |
Included |
|
Lines 110-115: Good. The rest of the paper focuses on semantic density. On the first read, I wanted an example of semantic gravity. But on a second read, I think you’d be OK without it. But if another reviewer would like it, go for it. |
Another reviewer did make this comment, so I reframed this section to include more detail on LCT and Semantics, including semantic gravity. |
|
Section 2/Lines 125-141: Can you please clarify that these are undergraduate/graduate/both? Without seeing the previous paper, it is useful to know a bit about the demographics of the students providing these products. |
Clarified. |
|
Around Table 2: I wondered how you defined a “clause”. Is this just the standard language definition of a clause? Or did you use a different definition? Clarification on this would help the reader. |
This is the standard definition of the clause. Additional information on the role linguistics played in the analysis has been added to the final paragraph of the introduction. |
|
Lines 237-238: Please clarify if it is LCT & physics, or physics & science |
clarified |
|
Figure 3: so this means there was no simple text image? Please clarify that I am reading this right. |
Clarified in the text (I excluded this in the Figure but did not justify this in the paper- this has now been added) |
|
Figure 1 is initially described as the semantic density per clause: the textual analysis. Then in line 288 it suggests that Figure 1 is for the blended media (text and image). Do you need two figures? |
Figure 1 represents the semantic density of the narration only. Points A1, A2, B1 and A3 on Figure 1 were then identified as ‘points of interest’, and detailed analysis including both image and text were conducted at these points. Changes in the results section have been made to clarify this. |
|
In section 4.2, lines 499-517 feels too detailed. Maybe put some of this in the introduction, or simplify it for the reader who is a novice to LCT? |
Some of the text on the ‘translation device’ has been moved to the Introduction to clarify this.
|
|
The discussion could use a little more discussion of how teachers could use this. It currently has some, but at the more theoretical level. It would be nice to see a clear “we plan to make a tool for marking digital products” or some other statement to give teachers a reason to keep up with this research. |
Clarification/elaboration has been added to Implications and limitations sections. |
|
|
|
Reviewer 4 Report
Please use the following to strengthen your manuscript:
Introduction
Please define constructs like ICT, tertiary level education and assessment for the reader from the relevant literature (rather than just providing an in-text reference). It helps the reader get up to speed faster. The sentence in lines 31-33 need to be split up and clarified as these are 2 separate thoughts. "There is sufficient evidence to suggest using DSPs as assessment tasks may be advantageous for learners in terms of developing disciplinary knowledge, increasing engagement, facilitating collaborative skills" -assessments as in tests? How are students developing knowledge during tests (assessments)? Please give examples from 12, 13, and 14 in text rather than having the reader to look it up. The Dynamic Standalone Product (DSP) needs a bit more defining and detail given it is a central part of the manuscript. Just 2-3 sentences would suffice. Line 45 that starts with "There, a robust corpus establishes..." is a summary statement, linking together literature and should be the last sentence of the paragraph. The literature in this 3rd paragraph should reflect tertiary level education, not primary and 9th grade. Literature from tertiary level should be referenced and discussed instead. Cognitive Load Theory should emphasize design elements made for the end user, that will make the transition towards discussing content made by the user stronger. What is meant by novices in line 78? Clarification is needed to understand the user generated content here. 108 should be and instead of & LCT consists of five dimensions, each focused on a different element of practices known as its organising principles - briefly list them ending with semantic density (the subconstruct of interest). The last paragraph should conclude with a statement pivoting back to user made content to better set up the reader for the materials and methods section.
Materials and Methods:
Is the non-specialist audience the same as tertiary students, like by students and for students? Discussion of why these 2 sample texts were selected is warranted. Sample size limitation should be acknowledged, as contributing to a preliminary conceptual framework. Move the IRB statement to the end of the manuscript per the author guidelines.
Analysis:
Trustworthiness is an important element of content analysis research and should be discussed in detail to ensure good faith of results.
Discussion:
Line 399-400 states that "In this paper, we aimed to understand how to assess student understanding of science and communication..." Is it really student understanding or rather, how analyze the level of understanding (of science and communication...) presented in digital content created by students? The latter seems within the purview of the paper (particularly the purpose (need/gap) and analysis) and then it could be stated that through this analysis is one means of (by proxy) to gain insight to student understanding of the rules of game in relation to understanding science. Meaning, that pays better homage/aligns to the nature of the theory as stated in lines 82. Line 412, there should be an and instead of & Line 518, interrater or intercoder reliability should be discussed in detail and placed in the trustworthiness section of the analysis. Lines 522 and 523, many scholarly traditions eschew use of rhetorical questions. These should be changed to align to best practices.
Conclusions:
Overall, the conclusion seems weak despite the good level of content presented in the manuscript. Therefore, in a paragraph, summarize both salient results and best practice recommendations of the study. Describe some avenues for future research.
Author Response
Thank you to Reviewer 4. Each issue raised is addressed in the table below and amended manuscript.
|
Please define constructs like ICT, tertiary level education and assessment for the reader from the relevant literature (rather than just providing an in-text reference) |
ICTs is an acronym for Information and Communications Technologies. Examples are provided. Tertiary level has also been clarified. |
|
The sentence in lines 31-33 need to be split up and clarified as these are 2 separate thoughts. "There is sufficient evidence to suggest using DSPs as assessment tasks may be advantageous for learners in terms of developing disciplinary knowledge, increasing engagement, facilitating collaborative skills" -assessments as in tests? How are students developing knowledge during tests (assessments)? |
I have removed ‘assessments’ in order to improve clarity here. Students learn the content and a range of other skills as part of engagement in the construction process of a DSP. Relevant literature has been included. |
|
Please give examples from 12, 13, and 14 in text rather than having the reader to look it up. |
Examples provided. |
|
The Dynamic Standalone Product (DSP) needs a bit more defining and detail given it is a central part of the manuscript. Just 2-3 sentences would suffice. |
Added. The products themselves will also be provided as supplementary materials |
|
Line 45 that starts with "There, a robust corpus establishes..." is a summary statement, linking together literature and should be the last sentence of the paragraph. |
This has been amended |
|
The literature in this 3rd paragraph should reflect tertiary level education, not primary and 9th grade. Literature from tertiary level should be referenced and discussed instead. |
This has been added to paragraph 2 and paragraph 3 has been reworded in order to justify this discussion. |
|
Cognitive Load Theory should emphasize design elements made for the end user, that will make the transition towards discussing content made by the user stronger. |
Noted. |
|
What is meant by novices in line 78? Clarification is needed to understand the user generated content here. |
Amended |
|
108 should be and instead of & LCT consists of five dimensions, each focused on a different element of practices known as its organising principles - briefly list them ending with semantic density (the subconstruct of interest). |
Amended. |
|
The last paragraph should conclude with a statement pivoting back to user made content to better set up the reader for the materials and methods section. |
Added |
|
Is the non-specialist audience the same as tertiary students, like by students and for students? Discussion of why these 2 sample texts were selected is warranted. Sample size limitation should be acknowledged, as contributing to a preliminary conceptual framework. Move the IRB statement to the end of the manuscript per the author guidelines. |
Audience clarified. The two texts were selected as they represented lower/lowest and highest marks (4, and 5). Sample size limitation further discussed in the relevant section.
|
|
Trustworthiness is an important element of content analysis research and should be discussed in detail to ensure good faith of results. |
Reliability discussed further in methods section. |
|
Line 399-400 states that "In this paper, we aimed to understand how to assess student understanding of science and communication..." Is it really student understanding or rather, how analyze the level of understanding (of science and communication...) presented in digital content created by students? The latter seems within the purview of the paper (particularly the purpose (need/gap) and analysis) and then it could be stated that through this analysis is one means of (by proxy) to gain insight to student understanding of the rules of game in relation to understanding science. Meaning, that pays better homage/aligns to the nature of the theory as stated in lines 82 |
Thanks for this suggestion!
|
|
Line 412, there should be an and instead of & Line 518, interrater or intercoder reliability should be discussed in detail and placed in the trustworthiness section of the analysis. |
All ampersands changed to ‘and’
|
|
Lines 522 and 523, many scholarly traditions eschew use of rhetorical questions. These should be changed to align to best practices. |
Rhetorical questions removed. |
|
Overall, the conclusion seems weak despite the good level of content presented in the manuscript. Therefore, in a paragraph, summarize both salient results and best practice recommendations of the study. Describe some avenues for future research. |
Done. Thank you. |
Round 2
Reviewer 2 Report
Thank you for attending to the comments on the manuscript. Although there are parts where further examples would be beneficial, overall it is a well thought out work.